# Benefits of Telemonitoring of Pulmonary Function—3-Month Follow-Up of Home Electronic Spirometry in Patients with Duchenne Muscular Dystrophy

**DOI:** 10.3390/jcm11030856

**Published:** 2022-02-06

**Authors:** Eliza Wasilewska, Agnieszka Sobierajska-Rek, Sylwia Małgorzewicz, Mateusz Soliński, Ewa Jassem

**Affiliations:** 1Department of Allergology and Pulmonology, Medical University of Gdańsk, 80-211 Gdańsk, Poland; ejassem@gumed.edu.pl; 2Department of Rehabilitation Medicine, Medical University of Gdańsk, 80-211 Gdańsk, Poland; sobierajska@gumed.edu.pl; 3Department of Clinical Nutrition, Medical University of Gdańsk, 80-211 Gdańsk, Poland; sylwiam@gumed.edu.pl; 4Faculty of Physics, Warsaw University of Technology, 00-661 Warsaw, Poland; mateusz.solinski.dokt@pw.edu.pl

**Keywords:** telemonitoring, digital health, home monitoring, e-monitoring of pulmonary function, home monitoring pulmonary function, rare diseases, Duchenne muscular dystrophy, pulmonary function test, spirometry, AioCare, COVID-19, adherence

## Abstract

Background: In patients with Duchenne Muscular Dystrophy (DMD), the respiratory system determines the quality and length of life; therefore, the search for easy and safe everyday monitoring of the pulmonary function is currently extremely important, particularly in the COVID-19 pandemic. The aim of the study was to evaluate the influence of a three-month home electronic spirometry (e-spirometry) monitoring of the pulmonary function and strength of respiratory muscles as well as the patients’ benefits from this telemetric program. Methods: Twenty-one boys with DMD (aged 7–22; non-ambulatory-11) received a remote electronic spirometer for home use with a special application dedicated for patients and connected with a doctor platform. Control of the hospital spirometry (forced vital capacity-FVC, forced expiratory volume in 1 second-FEV1, peak expiratory flow-PEF) and respiratory muscle strength (maximal inspiratory-MIP and expiratory pressures-MEP) before and after the three-month monitoring were performed as well telemonitoring benefit survey. Results: A total of 1403 measurements were performed; 15 of the participants were able to achieve correct attempts. There were no differences between the hospital and the home spirometry results as well as between respiratory muscle strength during v1 vs. v2 visits for the whole study group (all parameters *p* > 0.05); the six participants achieved increased value of FVC during the study period. There was a positive correlation between ΔFVC and the number of assessments during the home spirometry (r = 0.7, *p* < 0.001). Differences between FVC and MIP_cmH2O_ (r = 0.58; *p* = 0.01), MEP_cmH2O_ (r = 0.75; *p* < 0.001) was revealed. The mean general satisfaction rating of the telemonitoring was 4.46/5 (SD 0.66) after one month and 4.91/5 (SD 0.28) after three months. The most reported benefit of the home monitoring was the improvement in breathing (38% of participants after one month, 52% after three months of telemonitoring). Forgetting about the procedures was the most common reason for irregular measurements; the participants reported also increased motivation but less time to perform tests. Conclusions: The study indicates high compliance of the home telemonitoring results with the examination in the hospital. Benefits from home spirometry were visible for all participants; the most important benefit was breathing improvement. The remote home spirometry is usable for everyday monitoring of the pulmonary function in DMD patients as well can be also treated as respiratory muscle training.

## 1. Introduction

Duchenne Muscular Dystrophy (DMD) is the most common and progressive muscular dystrophy in childhood. Respiratory system and heart functioning determine the patient’s life expectancy, which, thanks to multi-specialized coordinated care, has now increased from a dozen to around 30 years [1].

Respiratory muscle weakness seems to be a major component of respiratory dysfunction [2]. It can be observed that the consequence of these changes is the reduction of the vital capacity of the lungs. It is known that pulmonary function declines at a rate of 6–11% annually in patients with DMD [3,4,5]. Moreover, respiratory muscle weakness leads to secondary changes such as decreased lung compliance, ineffective cough with deterioration of airway clearance, and repeated infections [6,7].

The time of the SARS-Cov-2 virus pandemic caused a search for new solutions in the field of diagnosis, monitoring, and contact with the patient and the acceleration of their implementation. The methods of telemonitoring of the patients’ condition at home are known and available in some specialties, e.g., cardiology, endocrinology, psychiatry, and geriatrics [8,9,10,11,12]. Automatic transmission of physiologic data (weight, blood pressure, heart rate and rhythm) using networked devices through mobile phones, the transmission of data acquired from wearable monitoring technology (heart, respiratory rate and temperature), or implanted therapeutic devices (pacemakers, defibrillators) are widely used in cardiology care [13,14]. Remote monitoring devices transmitting blood glucose levels, blood pressure, heart rate and weight are useful in everyday practices in patients with diabetes mellitus [15,16]. There are some reports of telemonitoring in pulmonology patients with asthma, cystic fibrosis, and after lung transplants; however, there are still no appropriate telemetric methods for monitoring lung function in patients with neuromuscular disorders [17,18].

In patients with DMD, the respiratory system determines the quality and length of life; therefore, the search for easy and safe monitoring of the pulmonary function is currently extremely important, particularly in the COVID-19 pandemic, when personal contact of the patient with the physician is limited [19,20]. Clinicians need to know the patient’s clinical condition, especially in the case of chronic disease. Moreover, methods of improving the functioning of the muscles of the respiratory system are also limited. Steroid therapy (prednisone or deflazacort) is a confirmed factor that delays the deterioration of lung function but does not cause its’ improvement [21]. In turn, the DELOS study, (Duchenne muscular dystrophy long-term idebenone study) showed that idebenone, a short-chain benzoquinone, significantly reduced the loss of pulmonary function over the 52-week study period in a DMD patient cohort not using a concomitant corticosteroid therapy. Currently, this treatment is only available in clinical trials [22,23,24]. Respiratory training is recommended as one of the key elements for DMD patients and is another method of improving pulmonary function [25,26,27,28]. Currently, during the COVID-19 pandemic, preference is given to the telerehabilitation of respiratory muscles. However, it was found that some patients had difficulties with telerehabilitation such as remembering to do the exercises at all or problems with the regularity of training at home [29]. It became necessary to find a solution for the problem of monitoring the respiratory system during exercises and reminding patients about performing the exercises.

That is why we implemented a program for remote everyday monitoring of pulmonary function in DMD patients. In our previous study, we showed that it was possible to measure and monitor lung function at home by the patients themselves [30]. Currently, we wanted to evaluate what is the effect of using this method for the next three months on lung function in these patients.

The aim of the study was to evaluate the influence of the three-month home e-spirometry monitoring on the pulmonary function and strength of respiratory muscles as well as the patients’ benefits from this telemetric program.

## 2. Materials and Methods

### 2.1. Study Design

This prospective, control-case, open-label study was conducted from June 2021 to October 2021 as the third part of the project: “E-monitoring of the pulmonary function in patients with Duchenne muscular dystrophy undergoing respiratory rehabilitation at home”; details of the other parts were described elsewhere [30].

The patients with DMD were recruited from Rare Disease Centre, Medical University of Gdańsk, University Clinical Centre; Poland which is a member of the TREAT NMD Alliance Neuromuscular Network. Approval for the study was obtained from The Local Committee of Ethics no. NKBBN/260/2021, which conformed to the principles embodied in the Declaration of Helsinki. Written informed consent was obtained from the participants and parents of each patient.

### 2.2. Participants

The study population included boys with DMD diagnosis based on the guidelines [1]. Inclusion criteria were as follows: (1) Male, ≥7 years and <25 years of age; (2) ability to perform spirometry and respiratory muscle strength test; (3) stated willingness to comply with all study procedures (AioCare device used) and availability for the duration of the study; and (4) no COVID-19 infection.

Due to the number of telemonitoring devices, 30 boys who met the criteria were invited to participate in the project. The duration of the study was 12 weeks, during which the participants made 3 visits at hospital.

V1 visit (enrolment): The physical examination, anthropometry measurement, Vignos scale (VS), Brooke scale (BS), handgrip strength and respiratory system assessment: spirometry (Jaeger, Hoechberg, Germany), and respiratory muscle strength test (MicropPRM, Rochester, Kent, UK) were performed in all enrolled patients. Each patient received a device for telemonitoring: electronic individual spirometer (AioCare) for daily spirometry measurements at home.

V2 visit (control): After 4 weeks of telemonitoring, patients came to evaluate the correctness of the e-spirometry measurements. They also filled in a questionnaire: Telemonitoring benefits survey (survey 1).

V3 visit (follow up): At the follow-up after 12 weeks, hospital spirometry, respiratory muscle strength tests were conducted and the Telemonitoring benefits survey (survey 2) was filed in by participants again.

### 2.3. Respiratory System Assessment

#### 2.3.1. Spirometry (Telemonitoring and Hospital Spirometry Control)

Telemonitoring of pulmonary function was performed by AioCare system, which consists of the device for home spirometry (AioCare^®^ spirometers, Healthup, Warsaw, Poland), AioCare^®^ Patient application, and AioCare^®^ Doctor platform [31]. AioCare e-spirometer is a small, convenient device and can be used anywhere; even by children over 5 years of age. The participants were asked to perform on their own three correct spirometry measurements twice daily, morning and evening at home during a 12-week period. They performed measurements themselves with the help of caregivers after a previous 4-week training period of using AioCare device at home, which was described in another publication [30].

The hospital control spirometry (Pneumo Screen, Jaeger, Hoechberg, Germany spirometer) the participants performed twice (during the V1 and V3 visit) by the same pediatric pulmonologist who evaluated these results. A minimum of 3 and up to 5 manoeuvres with a maximum effort were attempted by each participant.

All measurements of the hospital and the home spirometry were performed according to the European Respiratory Society and American Thoracic Society recommendations [32,33]. The highest value of forced vital capacity (FVC), forced expiratory volume in 1 s (FEV1), peak expiratory flow (PEF) expressed as litres (L), litres per minute (L/min) accordingly, and percent predicted value from correct acceptable attempts were evaluated.

The results of hospital spirometry were submitted to the ePULMoDMD database. The test results of the home e-spirometry were sent from the AioCare spirometer via the AioCare application for iOS or Android (used by participants’ smartphones) to the AIOCARE Doctor on-line panel and were available to the practitioner in real time. Finally, all test results of the home e-spirometry were also collected in the ePULMoDMD Database. The results were compared with results from the hospital spirometry.

#### 2.3.2. Respiratory Muscle Assessment

Respiratory muscle strength was assessed by measuring the maximal inspiratory pressure (MIP), and the maximal expiratory pressure (MEP) using a mouth pressure meter (MicroRPM; Micro Medical Ltd., Rochester, England). Operators were specially trained to do MIP and MEP. The manoeuvre made by the subject was a maximal inspiration or expiration sustained for at least one second against a blocked airway. The highest positive MEP value and the lowest negative MIP value in three or more attempts were chosen and calculated percent predicted MEP (MEP%) and MIP (MIP%) values according to the formulas: MEP × 100/(7.619 + (7.806 × age) + (0.004 × height × weight) and −MIP × 100/(–27.020 − (4.132 × age) − (0.003 × height × weight) [34].

#### 2.3.3. Functional Status and Hand Grip Strength

The severity of the disease was evaluated by ambulatory status (yes/no) and the Vignos scale (VS). The vs. allows staging of the disease and focuses on functional ambulatory activities. The scores on the vs. range from 1 to 10 (1—the subject can walk and climb stairs without assistance, 10—the subject is confined to a bed) [35].

Upper limb functional status was assessed with the 6-point Brooke scale (BS) (1—the subject can abduct their arms in a full circle until they touch above their head, 6—the subject has no useful function of the hands) [36].

Maximal grip strength was measured using dynamometer FT-5988-N1 (Spais, Gdansk, Poland). Trials were carried out with strong verbal encouragement, asking the patients to provide maximal voluntary isometric contractions for 3 s. The remaining period of about 30 s was respected between the trials. Patients were asked to perform three trials. The mean value was calculated from three valid trials [37].

### 2.4. Telemonitoring Benefits Survey

Details of satisfaction of patients and benefits from using the home telemonitoring of pulmonary function were collected from a survey after 1 month (survey 1) and after 3 months (survey 2) of using the home spirometry. The patients were asked to express their general satisfaction from the AioCare home spirometry on the 5-point scale, where “1” meant the worst and “5” meant the best score. The other questions concerned benefits and problems with regular measurements from the home spirometry.

### 2.5. Statistical Analysis

The results of the statistical analysis were expressed as mean (standard deviation) or median (interquartile range). The comparison of the mean values of the spirometry parameters between the averaged home spirometry results (from 3 months of monitoring) and ambulatory spirometry was performed using paired, parametric Student *t*-test, or non-parametric Wilcoxon signed-rank test (depending on whether the compared values were normally distributed, which was evaluated using the Shapiro–Wilk test) with a significance level of *p* = 0.05. Pearson correlation coefficients between spirometry parameters obtained during either the home and the laboratory spirometry and other clinical indices were calculated. The relationship between the changes of FVC (and FVC%) parameter during the home monitoring as well as between ambulatory visits and other clinical data (age, BMI, handgrip strength, MIP, MEP, ambulatory status, and number of spirometry measurements) was analyzed using backward stepwise linear regression models. Changes of the spirometry parameters values (Δ) over 90 days were calculated using linear regression for each patient; the Δ value was defined as the difference of the estimated value of the parameter from the linear regression model between the 90th and the 1st day of monitoring.

## 3. Results

### 3.1. Participants

Finally, 21 out of 30 participants were notified of the v1 visit and received an e-home spirometer. The characteristic of the study group is presented in Table 1.

### 3.2. Pulmonary Function Test

#### 3.2.1. Telemonitoring Compliance

Using home e-spirometry, the participants performed a total of 1403 measurements (mean 66.8 (SD 49.8) over 90 days of study time. Fifteen patients were able to perform correct attempts according to ATS/ERS 2019 criteria [33]. Sessions with at least one correct measurement presented 85.5% of the participants. At least 30 days with one performed assessment per day presented thirteen participants and with at least two assessments per day-six patients (Table 2). Since two participants took less than 10 measurements during the study period, they were excluded from further calculations (ID3 patient, ID15 patient).

In the final study group (*n* = 19) the correctness of spirometry increased, although the difference between 1 and 2 + 3 months was not statistically significant both for examinations meeting ATS/ERS 2019 criteria (25.4% vs. 37.8%, *p* = 0.130) and for examinations with at least one correct manoeuvre (81.6% vs. 88.2%, *p* = 0.173)—see Figure 1A,B.

The compliance analysis showed that the mean adherence decreased significantly in two and three months in comparison to the first month (30.1% vs. 18.3%; *p* = 0.019 for days with two examinations per day) and (63.2% vs. 46.7%; *p* = 0.0022 for days with one examination per day)—see Figure 1C,D.

#### 3.2.2. Study Group Spirometry

The mean values of the home telemonitoring and hospital spirometry are presented in Table 3. There were no differences between mean hospital vs. home parameters of spirometry (FVC, FEV1, PEF) (for all parameters *p* > 0.05). There were also no differences in mean parameters of the hospital spirometry between the first and third visits as well as between day one and day 90 in the mean value of the home e-spirometry (ΔFVC, ΔFEV1, ΔPEF) (for all parameters *p* > 0.05); (Table 3).

#### 3.2.3. Individual Spirometry

For each individual participant, the value of FVC measurement during the hospital and the home spirometry is presented in Table 4. Change in the home e-spirometry parameter values (ΔFVC L and ΔFVC %pv) over 90 days was calculated using linear regression for each parameter. After fitting a straight line, the difference at d = 90 days (or the last day of monitoring) and day = 0 was calculated. Results of ΔFVC revealed increasing trends in the six selected patients (see Table 4 and Figure 2). Visualizations for other individual parameters of spirometry (FEV1, PEF) are presented in Appendix A. This subgroup performed over 65 measurements per patient during the study period and at least 50% of manoeuvres were correct. There was a positive correlation between ΔFVC (L) and the number of assessments through home spirometry (r = 0.7, *p* < 0.001); however, no relationship was revealed with other telemonitoring compliance (number of correct examinations of the home spirometry and adherence) for all participants (for all parameters *p* > 0.05).

#### 3.2.4. Respiratory Muscle Assessment

The value of maximal expiratory and inspiratory pressure are given in Table 5. There were no differences between the first and third visits for mean parameters of MIP (ΔMIP; *p* = 0.169), and MEP (ΔMEP; *p* = 0.533). There was also no correlation between MIP and MEP parameters and age (*p* > 0.05).

### 3.3. Relationship between FVC and Respiratory Muscles

Basing on the analysis of the assessments from v1 visit, the correlation between FVC (L) and MIP_cmH2O_ (r = 0.58; *p* = 0.01) as well between FVC (L) and MEP_cmH2O_ (r = 0.75; *p* < 0.001) was revealed. There was also a fair correlation between FVC% vs. MEP% (r = 0.46; *p* = 0.05) and no correlation between FVC% vs. MIP% (r = 0.32; *p* = 0.20)—see Figure 3A–D.

### 3.4. Relationship between FVC and Motor Function

Neither BS nor vs. was correlated with FVC or ΔFVC measured both in the home and the hospital spirometry and with telemonitoring compliance (frequency, correctness, adherence) (for all parameters *p* > 0.05). There was correlation between handgrip strength and FVC (L) (r = 0.70; *p* < 0.001) as well ΔFVC measured in the home e-spirometry (r = 0.67; *p* < 0.001) and the hospital spirometry (r = 0.61; *p* = 0.02)—see Figure 3E,F.

### 3.5. Regression Analysis

Step-wise linear regression analysis showed that patients with better handgrip strength presented a higher ΔFVC% in the home e-spirometry (*p* = 0.016) (model 1) and also in the hospital spirometry (model 2, 3) (*p* = 0.002 and *p* = 0.001, respectively). Correlation coefficients between clinical variables and ΔFVC (L) are presented in Table 6.

### 3.6. Telemonitoring Benefits Survey

The ratings of general satisfaction varied from five to three for Survey One and from five to four for Survey Two. The mean general satisfaction rating of the home spirometry was 4.46/5 (SD 0.66) in Survey One and 4.91/5 (SD 0.28) in Survey Two. There was no significant difference between general satisfaction in Surveys One and Two (*p* = 0.095). Benefits from the home spirometry were visible for 80% of respondents in Survey One and all respondents in Survey Two. Breathing improvement was indicated as the most important benefit from the telemonitoring pulmonary function for more respondents (Figure 4). The most common reason for irregular measurements was forgetting about the procedures. The patients reported increased motivation but less time to perform tests. Reasons for benefits and irregular measurements reported by the participants in Surveys One and Two are presented in Figure 4.

## 4. Discussion

The present study looked at the clinical effects of telemonitoring of lung function by daily use of a home handheld e-spirometer-networked device with automatic transmission through mobile phone results of pulmonary function tests in patients with DMD.

Our study confirmed high compliance of the home telemonitoring assessments with the examinations performed in the hospital. Moreover, the telemonitoring results showed that all patients were able to perform the tests correctly at home, although the ATS/ERS2019 correctness criteria were not reached by 20% of the patients.

It should be taken into account that the correctness criteria of spirometry published by ATS/ERS2019 are very demanding. Repeatability criteria applied acceptance to FVC require three correct manoeuvrers and the difference between the two highest FVC values is less than 150 mL [33]. Patients with DMD, due to muscle weakness, may have difficulties achieving all of the above-listed repeatability criteria in subsequent measurements; however, they are able to perform at least one correct measurement during a session. Despite this, spirometry is still recommended to monitor lung function in patients with DMD [2].

The results of our patients’ spirometry are surprisingly good in comparison to the study by Eaton et al., where, after attending spirometry workshops, only 33% of tests performed by trained primary care technicians met the guidelines of ATS criteria [38]. In turn, the other studies showed that lung function test results collected during the home asthma telemonitoring are comparable to those collected under the supervision of trained professionals on commercial spirometers [35,39].

Attempts to conduct telespirometry have so far been used for obstructive diseases such as asthma and chronic obstructive lung disease (COPD) around 20 years previous. The most widely it was applied in patient self-monitoring of daily medication and in clinical drug efficacy trials. Most of them were simple peak flow meters, which showed only PEF (peak expiratory flow). In the 1990s, Abboud and Bruderman published articles about a transtelephonic portable personal spirometer (so-called Spirophone AG-SP) with a remote receiving centre (CG-80 10) which can measure lung function indices at a patient’s home and transmit the data to a remote receiving centre for the analysis of both spirograms and flow-volume curves. After the evaluation of 15 subjects, the authors concluded that the devices were suitable only for patients who were able to perform the forced expiratory manoeuvre by themselves and could respond to advice given over the telephone. In our AioCare system, the patients have real-time information about the correctness of the maneuvers performed, which allows us to correct their assessment [40,41].

At that time, for patients with asthma, the Home Asthma Telemonitoring (HAT) system was evaluated by Finkelstein, in which all seventeen patients, out of 19 invited, participated in the study and were monitored for three weeks. This system aimed to help to follow their self-care plans according to the NAEPP recommendations; however only 29.5% of patients independently reviewed their results at least once a week at home [17]. Moreover, in asthmatic children, portable spirometers were used, but initially only in clinical trials [42].

Gradually, spirometry was increasingly used for the assessment of lung function in children—not only in relation to asthma but also for other diseases such as cystic fibrosis (CF) and primary ciliary dyskinesia (PCD), and not only for clinical trials but also for daily monitoring of lung function [18,43].

Currently, about 16 devices for the monitoring of basic lung function parameters exist; however, only 31% provided in-app videos on how to perform the breathing manoeuvres. Only 44% of them give immediate feedback on the quality of the breathing manoeuvre. Information on the data security (63%), measurement accuracy (50%), and association with patient outcomes (0%) was commonly limited. Thus, the benefit from the portable electronic spirometers in asthma patients may be limited due to the lack of patient outcome data [44].

In our study, we found that all participants reported the benefits of home telemonitoring, and most of them pointed to an improvement in breathing as the most important benefit achieved. We tried to objectify these results through careful analysis of the difference in the spirometry between the first inclusive visit and the last one. In control hospital assessments, the study group did not achieve an improvement in the value of the spirometry parameters. However, analyzing the results of individual patients, some of them presented an increase in FVC, which was confirmed by control hospital tests. It is possible that this effect may be related to the interdependence of muscle strength and lung function, although, in our study, the strength of the respiratory muscles (measured as MIP, MEP) did not depend on the age of the patient but affected lung capacity. The participants with stronger respiratory muscles presented better lung capacity. The muscle strength of the upper limb also positively correlated with lung capacity, which resulted in higher forced volume capacity. This was confirmed by regression analysis, where the value of handgrip strength was the most important floating factor for FVC increase during the three-month study.

It also seems that the greatest benefit from e-spirometry was reached by patients who regularly took measurements. In the group regularly performing measurements, the correctness of measurements increased. Performing regular breathing manoeuvres during the home e-spirometry resulted not only in better patients experience in spirometry assessments but also in maintaining the strength of respiratory muscles. This could potentially lead to an increase in lung function in some patients.

In the literature, patients with other chronic diseases experienced measurable benefits of telemonitoring. The systematic reviews have found that telemonitoring for heart failure management reduces mortality risk and hospital readmissions and more frequent transmission of patient data increases its effectiveness [45]. Telemonitoring in diabetic patients (e.g., Telescot, REMOTE study) showed that this method results in an improvement in the control of blood glucose (BG) level and a significant reduction in HbA1c, with a positive impact on co-morbidities (arterial hypertension, weight, dyslipidemia) [15]. During the COVID-19 pandemic, telemonitoring has proven to be particularly useful and creating opportunities for continuity of care to patients in a complex and novel setting. During this time, in England, as many as 88% of 391 children with chronic respiratory diseases (CF, PCD, asthma) that received home telemonitoring devices felt it improved quality of care [46]. Amorim et al. found that the number of face-to-face consultations fell by 75% at the beginning of the pandemic and a portable home spirometer in patients with COPD proved to be very useful in ensuring continuity of care and the possibility of rapid therapeutic intervention in these patients [47].

Another interesting finding of our study was that telemonitoring seems to be most attractive to the patients in the first month of use, when regular measurements twice a day were used by over 30% of patients and single manoeuvres per day were performed by over 60% of participants. In the next two months of telemonitoring, the percentage of patients with good adherence fell to less than 20% (twice daily measurements) and 40% (single daily measurement), respectively. This form of telemonitoring is new for patients with DMD and further exploration of the method and outcomes are needed.

Limitations: The results we received should be interpreted with caution due to a small group with a wide age range and severity of the disease. All participants were only of Caucasian race. The bias was due to the fact that many factors can influence variability in pulmonary function tests, not only in patients with DMD but even in healthy subjects, which should be taken into consideration. Moreover, the normal variability overtime period of three months will also depend on the disease process being monitored.

Despite these limitations of the study, home telemonitoring of pulmonary function is an innovative method that facilitates the monitoring of the disease for both the patient and the doctor.

In our previous study, we showed that it is possible to make assessments of pulmonary function at home in patients with DMD [30]. The present study showed that lung monitoring in patients with neuromuscular diseases is not only possible but participants also benefit from using home spirometry electronic devices.

Knowledge about pulmonary functions in real-time is one of the benefits of remote everyday home e-monitoring. Some of the patients, by systematic breathing manoeuvres performed during the spirometry measurements, can experience an improvement in breathing, which in some cases was confirmed by an objective spirometry examination at the hospital.

## 5. Summary

Following the progress of patients with DMD and their lung disease is the question of how best to monitor their lung function. One of the innovative methods is the telemonitoring of lung function at home.

High compliance of the home telemonitoring and the hospital results of spirometry make this method usable for everyday use for monitoring lung function and exacerbations without having to come to the hospital, which is especially important during the COVID-19 pandemic.

The benefits of this method can be experienced by both the patients and the doctors. An improvement in breathing was the most frequently reported benefit by the study participants from the respiratory telemonitoring. Moreover, performing regular breathing manoeuvres during the home spirometry may result in improved general and physical well-being and conditioning.

It seems that the greatest benefit from e-spirometry is enjoyed by patients who regularly take measurements and have maintained the strength of respiratory muscle and handgrip strength.

Even if patients benefited from this method, the attractiveness and time spent on telemonitoring decreased over time. This may be due to the fact that the present form of telemonitoring requires activity on the part of the patient.

## 6. Conclusions

The study indicates high compliance of home telemonitoring results with examination in the hospital. Benefits from home spirometry were visible for all participants; the most important benefit was breathing improvement. Remote home spirometry is usable for everyday monitoring of the lung function in DMD patients and can also be treated as respiratory muscle training.

## Figures and Tables

**Figure 1 jcm-11-00856-f001:**
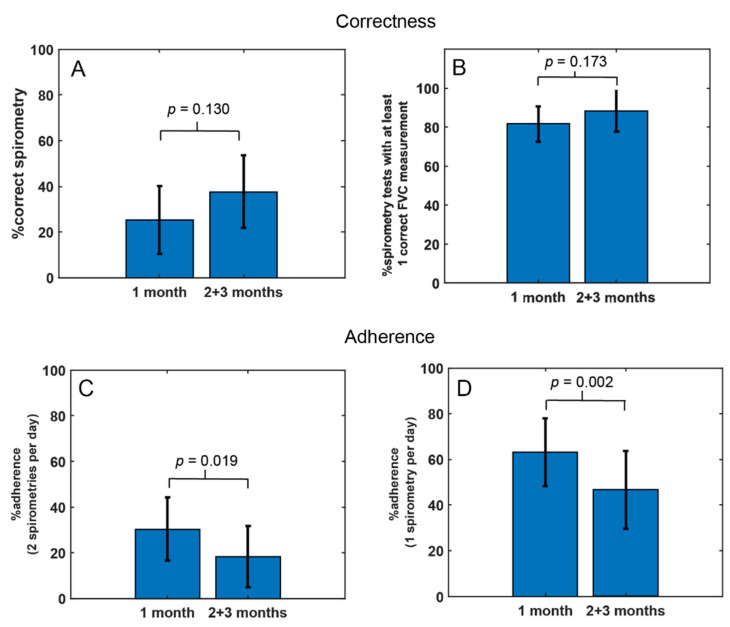
Compliance analysis of pulmonary function telemonitoring. Correctness: (**A**) percentage of the correct examinations (ATS/ERS 2019) [31]; (**B**) percentage of the exams with at least one corrected measurement. Adherence: (**C**) percentage of the days with at least two examinations/total number of the days ((30 or 60 days) × 100%); (**D**) percentage of the days with at least one measurement/total number of the days ((30 or 60 days) × 100%).

**Figure 2 jcm-11-00856-f002:**
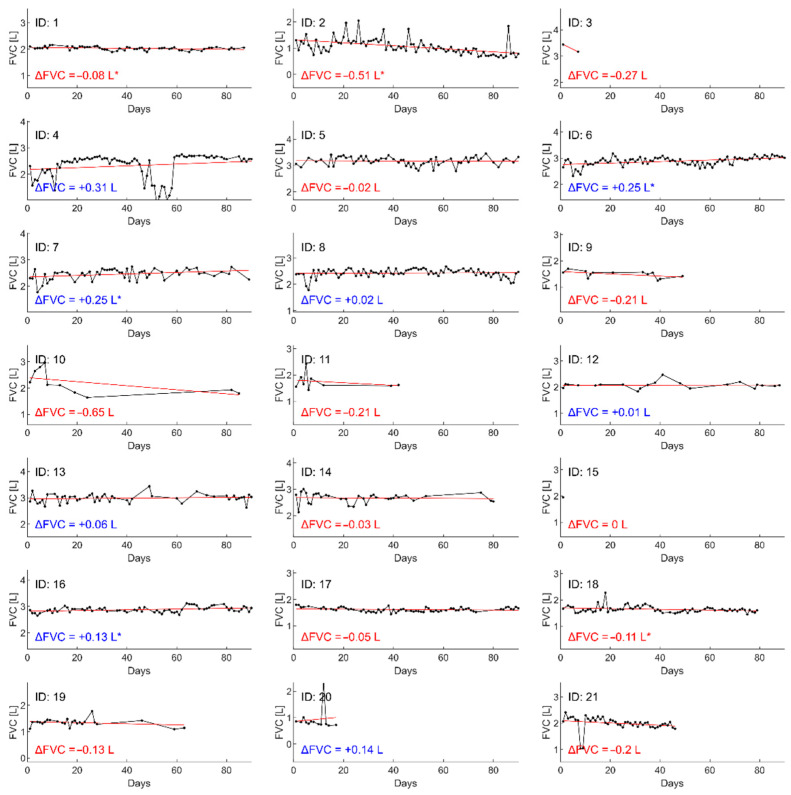
Daily forced vital capacity (FVC (L) charts with a matched straight line for individual participants. Change in spirometry parameter values (ΔFVC) over 90 days was calculated using linear regression for each parameter. After fitting a straight line, the difference at d = 90 days (or last day of monitoring) and day = 0 was calculated. Symbol * means that the slope coefficient of the fitted lines is significantly different from zero (*p*-value < 0.05).

**Figure 3 jcm-11-00856-f003:**
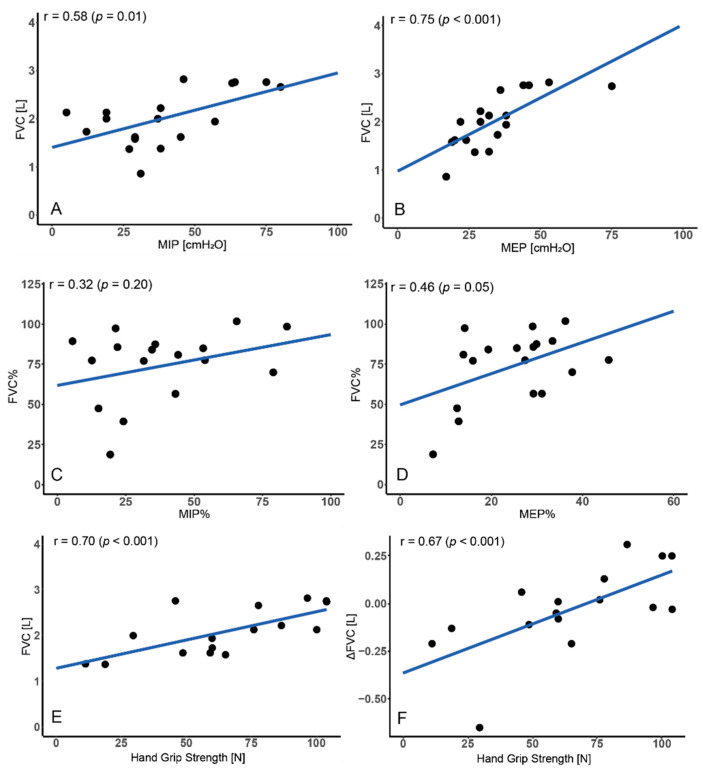
Relationship between forced vital capacity (FVC) and (**A**,**B**) maximal inspiratory pressure (MIP); (**C**,**D**) maximal expiratory pressure (MEP); (**E**,**F**) handgrip strength.

**Figure 4 jcm-11-00856-f004:**
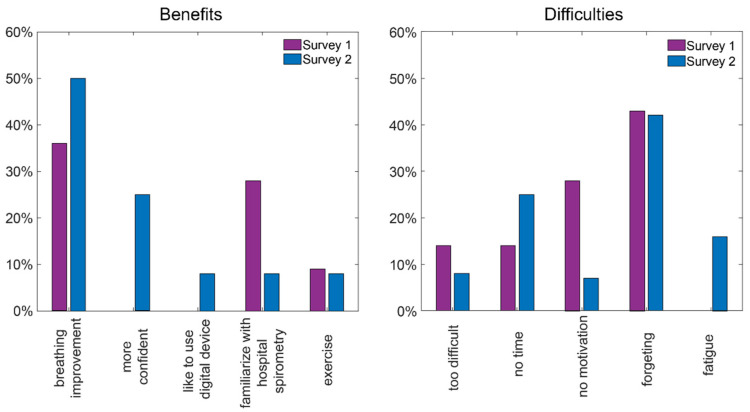
Benefits of telemonitoring of pulmonary function and reasons for irregular measurements reported by participants.

**Table 1 jcm-11-00856-t001:** Anthropometry and clinical characteristics of the study group.

	DMD Participants *n* = 21
Age years	12.8 (4.4)
Weight kg	51.3 (18.2)
Height cm	148 (16)
BMI kg/m^2^	22.7 (5.3)
Brooke scale	2 (4) Median (Q3–Q1)
Vignos scale	8 (7.5) Median (Q3–Q1)
Hand grip strength	66.3 (31.12)
Ambulatory yes/no	10/11
Mutation	Deletions 68.4% Point mutations 15.8% Duplications 10.5% Lack of mutation 5.3%

**Table 2 jcm-11-00856-t002:** Compliance with home e-spirometry for individual participants.

Participants ID	Number of Examinations	Correctness		Adherence	
		Correct Examinations (ATS/ERS 2019) nb (%)	Examinations with 1 Correct Maneuver nb (%)	Days with 2 Examinations nb (%)	Days with 1 Examination nb (%)
1	60	50 (83.3%)	58 (96.7%)	3 (3.3%)	57 (63.3%)
2	176	0 (0%)	47 (26.7%)	85 (94.4%)	89 (98.9%)
3	3	0 (0%)	1 (0%)	0 (0%)	2 (2.22%)
4	133	61 (45.9%)	114 (85.7%)	44 (48.9%)	82 (91.1%)
5	86	10 (11.6%)	65 (75.6%)	21 (23.3%)	64 (71.1%)
6	170	0 (0%)	165 (97.1%)	76 (84.4%)	90 (100%)
7	68	0 (0%)	39 (57.4%)	12 (13.3%)	55 (61.1%)
8	88	81 (92%)	88 (100%)	1 (1.1%)	87 (96.7%)
9	18	5 (27.8%)	18 (100%)	5 (5.6%)	13 (14.4%)
10	14	3 (21.4%)	11 (78.6%)	3 (3.3%)	10 (11.1%)
11	12	0 (0%)	11 (91.7%)	3 (3.3%)	9 (10%)
12	22	5 (22.7%)	20 (90.9%)	1 (1.1%)	21 (23.3%)
13	65	6 (9.2%)	57 (87.7%)	12 (13.3%)	50 (55.6%)
14	42	10 (23.8%)	27 (64.3%)	8 (8.9%)	34 (37.8%)
15	1	0 (0%)	1 (100%)	0 (0%)	1 (1.1%)
16	90	37 (41.1%)	89 (98.9%)	24 (26.7%)	66 (73.3%)
17	106	96 (90.6%)	105 (99.1%)	34 (37.8%)	69 (76.7%)
18	103	49 (47.6%)	84 (80.6%)	31 (34.4%)	71 (78.9%)
19	26	16 (61.5%)	26 (100%)	1 (1.6%)	25 (41%)
20	23	0 (0%)	21 (91.3%)	7 (21.2%)	13 (39.4%)
21	97	21 (21.6%)	78 (80.4%)	39 (45.9%)	46 (54.1%)
Total nb	1403	450	1125	410	954
Mean ± SD	66.8 (52.4)	21.4 ± 29.3 (28.6% ± 31.1%)	53.6 ± 43.1 (81.1% ± 25.8%)	19.5 ± 24.5 (22.5% ± 27.2%)	45.4 ± 30.7 (52.4% ± 32.9%)

nb, number.

**Table 3 jcm-11-00856-t003:** Mean values of hospital and home spirometry for study group (Mean ± SD).

	DMD Participants *n* = 19					
	Hospital		*p*-Value	Home Mean	*p*-Value	
	Visit 1	Visit 3			V1 vs. Home	V3 vs. Home
FVC (L)	2.02 (0.57)	2.24(0.55)	0.211	2.08 (0.66)	0.505	0.332
FVC (%pv)	74.0 (22.2)	80.1 (18.7)	0.307	74.2 (26.0)	0.918	0.905
FEV1/FVC	87.8 (7.6)	88.5 (6.2)	0.690	83.0 (18.3)	0.224	0.435
FEV1 (L)	1.82 (0.56)	2.00(0.48)	0.765	1.72 (0.71)	0.339	0.488
FEV1 (%pv)	77.6 (24.4)	85.8 (20.0)	0.141	71.3 (33.0)	0.170	0.230
PEF (L/min)	209 (56)	228 (55)	0.653	193 (80)	0.236	0.323
PEF (%pv)	69.0 (23.0)	74.2 (19.6)	0.606	64.26 (27.88)	0.322	0.371

forced vital capacity, FVC; forced expiratory volume in 1 s, FEV1; peak expiratory flow, PEF; pv, predicted value.

**Table 4 jcm-11-00856-t004:** Differences of forced vital capacity (FVC) values between two control points (start and 90 days follow-up) in hospital spirometry and home spirometry for individual participants.

Participants ID	Spirometry			
	Hospital		Home	
	ΔFVC (Liter)	ΔFVC (%pv)	ΔFVC (Liter)	ΔFVC (%pv)
1	0.12	5.4	−0.08 *	−3.09 *
2	−0.14	−13.8	−0.51 *	−30.36 *
3 ex	-	-	-	-
4	−0.04	4.3	0.31	10.36
5	0.24	7.7	−0.02	−0.55
6	−0.01	−0.8	0.25 *	8.82 *
7	0.4	16.2	0.25 *	9.82 *
8	0.13	5.2	0.02	0.71
9	−0.0	0.0-	−0.21	−10.43
10	−0.12	−2.9	−0.65	−15.28
11	−0.14	−5.5	−0.21	−9.3
12	−0.02	−1.1	0.01	0.4
13	−0.18	−4.4	0.06	2.78
14	0.07	2	−0.03	−0.91
15 ex	-	-	-	-
16	0.25	−2.7	0.13 *	4.98 *
17	−0.01	−0.5	−0.05	−2.55
18	0.12	0.0	−0.11 *	−3.37 *
19	−0.0	0.0-	−0.13	−8.47
20	−0.0	0.0-	0.14	2.95
21	−0.0	0.0-	−0.2	−3.74
Mean (STD)	0.04 (0.17)	0.61 (6.83)	−0.05 (0.24)	−2.36 (9.39)

ex, excluded participants; Differences for home spirometry were calculated as the value of FVC (L) or FVC% after 3 months minus estimated value at 1st day taken from linear regression model fitted to the data. Symbol * means that the slope coefficient of the fitted lines is significantly different from zero (*p*-value < 0.05).

**Table 5 jcm-11-00856-t005:** Values of respiratory muscle assessment.

	DMD Participants		*p*-Value V1 vs. V3
	Visit 1	Visit 3	
MIP_cmH2O_	39.7 (21.3)	49.4 (20.0)	0.169
MIP %pv	38.3 (22.3)	46.8 (24.2)	0.380
MEP_cmH2O_	34.2 (14.1)	40.4 (20.8)	0.533
MEP %pv	25.0 (10.5)	29.1 (17.6)	0.616

maximal inspiratory pressure, MIP; maximal expiratory pressure, MEP; predictive value, pv.

**Table 6 jcm-11-00856-t006:** Summary of stepwise linear regression models for predicting spirometry forced vital capacity (FVC (L) or ΔFVC%) or with the clinical, anthropometric, and spirometry data.

**Model 1 (ΔFVC%, e-Spirometry)**		
Variables	Coefficient (95%CI)	*p*-value
Intercept	−8.599 (−9.075~−8.409)	0.090
Age	−0.542 (−5.596~−0.539)	0.196
Hand Grip Strength	0.150 (0.149~0.154)	0.016
Number of measurements	0.062 (0.063~0.071)	0.078
**Model 2 (ΔFVC (L), hospital spirometry)**		
Variables	Coefficient (95%CI)	*p*-value
Intercept	0.092 (−0.117~0.485)	0.499
Hand Grip Strength	0.007 (0.011~0.018)	0.002
MEP	−0.005 (−0.017~−0.003)	0.117
BMI	−0.008 (−0.028~−0.003)	0.197
Number of measurements	−0.002 (−0.006~−0.002)	0.107
Ambulatory	−0.164 (−0.467~−0.190)	0.026
**Model 3 (ΔFVC%, hospital spirometry)**		
Intercept	4.632 (4.482~5.239)	0.237
Age	−0.403 (−0.450~−0.339)	0.131
Hand Grip Strength	−0.255 (−0.259~−0.268)	<0.001
MEP	−0.152 (−0.167~−0.151)	0.095
MIP	−0.117 (−0.121~−0.114)	0.010
Number of measurements	−0.065 (−0.070~−0.065)	0.030

## Data Availability

The data presented in this study are available on request from the corresponding author. Full data are not publicly available due to privacy restrictions.

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
