# Peer review of "Benefits of Telemonitoring of Pulmonary Function—3-Month Follow-Up of Home Electronic Spirometry in Patients with Duchenne Muscular Dystrophy"

_jcm, 2022, doi:10.3390/jcm11030856_

Round 1

Reviewer 1 Report

Telemonitoring of lung function is appealing in respiratory medicine especially during the pandemics of highly transmissible diseases. Daily, even twice daily measurement of FEV1 and FVC are very helpful for monitoring the disease with highly day-to-day variation or diurnal variation like asthma, or to the less extent, diseases which are prone to acute exacerbation, like COPD. In terms of DMD, a chronic, progressive neuromuscular disease, regular monitoring of FVC in combination with sleep studies, is reasonable and acceptable for patients. While everyday monitoring of spirometry might make patients boring and may not clinically be necessary.

This study confirmed high compliance of home telemonitoring of lung function in DMD patients. As the author said, it is very demanding to make the FVC maneuver fulfilling the criteria of ATS/ERS 2019 acceptability and repeatability, so it is not surprising that there are differences in lung function measurement at home or in hospital. It is not only the technician supervision and make timely orders, hand held device is not 100 percent comparable with flowsensors in PFT Lab which are calibrated and validated everyday. In this study most of the home data were not significantly different from those acquired in hospital, which was good enough for telemonitoring. But still there were opposite changes in FVC in some patients between home and hospital as shown in Table 4, which was understandable.

According to the author, the most impressive of this work is the improvement of breathing after three months twice daily spirometry measurement. To the best of our knowledge, even the regular pulmonary rehabilitation cannot improve lung function itself, but rather, improve the general and physical well-being and conditioning. It is hard to imagine that three months spirometry monitoring can raise FVC. Normal variation/fluctuation of measurement or growth and development cannot be excluded, a learning curve was also a possibility. The way the author calculate the change could be questionable, should be the difference of the mean, or the mean of the difference? Besides, can the small improvement in breathing fully attributed to the daily spirometry, or other factors existed just because of the lung function monitoring, like better nourishment or more physical movement?   

Because the small sample size, and short period of study relative to the duration of the DMD, it is better to be cautious to make the conclusion that DMD patients can benefit from telemonitoring in lung function improvement. And it is worth thinking of extending the observation, and making regular but less frequent monitoring in DMD patients to see if PFT measurement can really make a difference in lung function itself. It will keep the patients’ adherence to the monitoring till the disease progress, the day of ventilatory support.  

It will look much better if the author can make the paper less redundant.

Author Response

The Authors thank the Reviewer for a thorough evaluation of the manuscript. We have carefully addressed the comments of all Reviewers. The changes introduced in the text of the manuscript have been marked using the red font. Below, we present the detailed answers to the Reviewer’s comments and the description of the modifications introduced upon revision of the manuscript

This study confirmed high compliance of home telemonitoring of lung function in DMD patients. As the author said, it is very demanding to make the FVC maneuver fulfilling the criteria of ATS/ERS 2019 acceptability and repeatability, so it is not surprising that there are differences in lung function measurement at home or in hospital. It is not only the technician supervision and make timely orders, hand held device is not 100 percent comparable with flowsensors in PFT Lab which are calibrated and validated everyday. 

Author’s reply: We agree with the Reviewer that the difference between home and hospital spirometry is not surprising. However, the handheld spirometer AioCare used in the study does not require daily calibration due to thermal-based technology of the flow measurement in spite of laboratory spirometers (based on pneumotach technology) which need to be calibrated every day.

According to the author, the most impressive of this work is the improvement of breathing after three months twice daily spirometry measurement. To the best of our knowledge, even the regular pulmonary rehabilitation cannot improve lung function itself, but rather, improve the general and physical well-being and conditioning. It is hard to imagine that three months spirometry monitoring can raise FVC. Normal variation/fluctuation of measurement or growth and development cannot be excluded, a learning curve was also a possibility. 

Author’s reply: We changed according to the Reviewer's suggestion.

The way the author calculate the change could be questionable, should be the difference of the mean, or the mean of the difference? Besides, can the small improvement in breathing fully attributed to the daily spirometry, or other factors existed just because of the lung function monitoring, like better nourishment or more physical movement?   

Authors’ reply: In table 4 we calculated the mean of the differences instead of the differences of the mean. We assumed that FVC values (either for hospital and home spirometry) at visit 1 and 3 (in a case of hospital spirometry) or at 1st and 90th day (in a case of home spirometry) came from dependent groups which gave us an indication to analyze data in pairs by calculating the differences for each participants first.

Because the small sample size, and short period of study relative to the duration of the DMD, it is better to be cautious to make the conclusion that DMD patients can benefit from telemonitoring in lung function improvement. And it is worth thinking of extending the observation, and making regular but less frequent monitoring in DMD patients to see if PFT measurement can really make a difference in lung function itself. It will keep the patients’ adherence to the monitoring till the disease progress, the day of ventilatory support.  

It will look much better if the author can make the paper less redundant

We agree with the Reviewer, we have changed the conclusions to more cautious ones. We plan further observation as the Reviewer suggests.

Reviewer 2 Report

The present study is very interesting because it highlights the importance of home monitoring of lung function in patients with DMD.
However, some fundamental points emerged to be clarified for the purpose of publication and to improve a broader scientific response.
In particular these are major concerns:
1) In the protocol, why was such a tight monitoring time chosen?
2) I think that the lung function has improved considerably in such a short time is a bit risky. It would be possible to question that there is more of a tendency to improve, especially if we consider such a small number of patient trucks.
3) Another limitation is to be referred to the fact that the characteristics of the patients are not very clear; or did they all have the same degree of disability? Were the conditions of the DMD uniform? Please clarify these points.
4) The text highlights the importance of pulmonary rehabilitation in these patients. The authors define that even the repeated spirometric maneuvers have been found to be useful in rehabilitating the patient and for this reason the pulmonary function is improved. Is it scientifically plausible to confirm this? Are there other data in scientific literature to support it? Please insert some references. All this appears unclear because the duration of the study is very short and considering the spirometric maneuvers alone does not seem to be acceptable. Clarify even if the patients followed a parallel path of rehabilitation

Minor Concers:
1) Insert l'acronym DMD line 44 in the Introduction
2) Insert these references in the introduction that clarify the methods used in the hospital during the pandemic to safely perform pulmonary function tests: a)Practical considerations for spirometry during the COVID-19 outbreak: Literature review and insights. Pulmonology. 2021 Sep-Oct;27(5):438-447. doi: 10.1016/j.pulmoe.2020.07.011. Epub 2020 Aug 5. PMID: 32800783; PMCID: PMC7405879. b) Resumption of respiratory outpatient services in the COVID-19 era: Experience from Southern Italy. Am J Infect Control. 2020 Sep;48(9):1087-1089. doi: 10.1016/j.ajic.2020.06.210. Epub 2020 Jul 2. PMID: 32621858; PMCID: PMC7329653.

Author Response

The Authors thank the Reviewer for a thorough evaluation of the manuscript. We have carefully addressed the comments of all Reviewers. The changes introduced in the text of the manuscript have been marked using the red font. Below, we present the detailed answers to the Reviewer’s comments and the description of the modifications introduced upon revision of the manuscript

In particular these are major concerns:
1) In the protocol, why was such a tight monitoring time chosen?

Author’s reply: The study is scheduled for 12 months. Currently, we are publishing the first conclusions after a 3-month study period.

2) I think that the lung function has improved considerably in such a short time is a bit risky. It would be possible to question that there is more of a tendency to improve, especially if we consider such small number of patient trucks.

Author’s reply: We changed according to the Reviewer's suggestion.

3) Another limitation is to be referred to the fact that the characteristics of the patients are not very clear; or did they all have the same degree of disability? Were the conditions of the DMD uniform?

Author’s reply: Functional status was assessed with the use of Brooke scale (to assess upper limb functional status) and Vignos scale (to define level of ambulatory activities) as was presented in section 2.4.3. We presented the group characteristics in table 1, where it was stated that 10 from 21 participants were ambulant. Results of Brooke scale ranged from 1 to 6 and Vignos scale from 1 to 9, which reflects the high heterogeneity of the study population. However, our aim was not to compare particular patients to each other, thus to monitor change over time in patients in variable functional condition. Small sample size does not let us divide patients into groups according to the disease severity and indicate which group benefits most.

4) The text highlights the importance of pulmonary rehabilitation in these patients. The authors define that even the repeated spirometric maneuvers have been found to be useful in rehabilitating the patient and for this reason the pulmonary function is improved. Is it scientifically plausible to confirm this? Are there other data in scientific literature to support it? Please insert some references. All this appears unclear because the duration of the study is very short and considering the spirometric maneuvers alone does not seem to be acceptable. Clarify even if the patients followed a parallel path of rehabilitation

Author’s reply: Thank you for your attentive reflections. In the era of Covid, we have tried to introduce telerehabilitation at home in patients with DMD. However, this form was not very attractive to the patients and they mostly gave up breathing exercises.

It was a surprise to us that patients using AIOCARE daily reported improved breathing, the ability to do deeper breathing maneuvers, and overall an improvement in well-being in the survey. We also observed this on a follow-up examination after 3 months in the hospital.

These are novel studies and we do not find similar ones in the literature.  However, we believe that our research can inspire other researchers.

Minor Concers:
1) Insert l'acronym DMD line 44 in the Introduction
2) Insert these references in the introduction that clarify the methods used in the hospital during the pandemic to safely perform pulmonary function tests: a)Practical considerations for spirometry during the COVID-19 outbreak: Literature review and insights. Pulmonology. 2021 Sep-Oct;27(5):438-447. doi: 10.1016/j.pulmoe.2020.07.011. Epub 2020 Aug 5. PMID: 32800783; PMCID: PMC7405879. b) Resumption of respiratory outpatient services in the COVID-19 era: Experience from Southern Italy. Am J Infect Control. 2020 Sep;48(9):1087-1089. doi: 10.1016/j.ajic.2020.06.210. Epub 2020 Jul 2. PMID: 32621858; PMCID: PMC7329653

Author’s reply: We changed according to the Reviewer's suggestion.